# Triggers and Obstacles to the Development of the FinTech Sector in Poland

Agata Kliber [1,*], Barbara Będowska-Sójka [2], Aleksandra Rutkowska [3] and Katarzyna Świerczyńska [4]

1   Department of Applied Mathematics, Poznań University of Economic and Business, al. Niepodległości 10, 61-875 Poznań, Poland
2   Department of Econometrics, Poznań University of Economic and Business, al. Niepodległości 10, 61-875 Poznań, Poland; barbara.bedowska-sojka@ue.poznan.pl
3   Department of Applied Mathematic, Poznań University of Economic and Business, al. Niepodległości 10, 61-875 Poznań, Poland; aleksandra.rutkowska@ue.poznan.pl
4   Department of Economic Journalism and Public Relations, Poznań University of Economic and Business, al. Niepodległości 10, 61-875 Poznań, Poland; katarzyna.swierczynska@ue.poznan.pl
*   Correspondence: agata.kliber@ue.poznan.pl; Tel.: +48-61-854-3875

**Abstract:** The article aims to show the opportunities for the formation of new FinTech startups in Poland and further development of the sector, as well as to identify the most critical threats. The study offers the descriptive and deductive analysis based on the literature review. The empirical part relies on the data from external databases as well as the dataset collected in a survey run among the FinTechs in Poland in January 2020. The paper reveals that Poland is a fast-growing FinTech market which satisfies various requirements such as the number of secure Internet servers, mobile telephone subscriptions, the available labor force, as well as growing tertiary education enrolment. The crucial obstacles to the development of the sector is the uncertainty about the availability of skilled workers in the future and the lack of proper legal regulations.

**Keywords:** FinTech; development; economic environment

## 1. Introduction

The term 'FinTech', which is an abbreviation from Financial Technology, refers to software and other modern technologies used by a business that provides automated and improved financial services.[1] They introduce new financial products and offer them through disruptive technologies. FinTech is a good example of how innovation is ahead of regulation (Liu et al. 2020). Such entities began to flourish in the 1990s, together with the rapid increase of the internet, e-commerce businesses, and digitalization of banking and financial services. Some argue that also the Global Financial Crisis in 2008, in which many people lost their trust in traditional banking systems, became a driver of the development of this sector (Eddie 2020).

The FinTech sector develops worldwide, but the pace and the direction of this development are different in each country. According to the "Global FinTech Report" of PWC (2019), almost half of all firms in both financial services and technology, media and comunications (TMT) have fully incorporated FinTech-based products and services into their strategic operating models. Furthermore, more than half of banks and capital-markets companies have added emerging technologies solutions into commercial banking and personal loans. According to Deloitte report (Deloitte 2019), FinTechs have entered a new phase of their evolution from a formidable competitor to a trusted partner.

When it comes to geographical segmentation, China seems to be the FinTech-leader (PWC 2019): more than three-fourths of all products and services are supported with

---

1   https://www.FinTechweekly.com/fintech-definition.

Insuretech, while two-thirds offer robo-advices. Brazil takes second place in the ranking. PWC report (PWC 2019) states that organisations in China and Brazil are more likely to fully embed FinTechs across their strategic operating model (58% and 55%, respectively), compared with organisations in the developed economies such as the U.S. and Germany (37% and 36%, respectively). This is mainly due to the legislation and regulations—in China, the regulators are keen to promote FinTech-related innovations, while the European FinTechs are struggling more with the legacy, complexity, and cost structures.

The report titled "New Financial Geographies of Asia" of Lai et al. (2020) shows that while the capital markets in Asia are fast-growing, the Western banks experience serious problems in the region. Asian investment banks have gained more importance not only on the continent but also globally. So far, Hong Kong, Singapore, and Tokyo have been considered well-established international financial centers. The rise of financial centers in China suggests that the global financial landscape may be significantly re-designed in the nearest future (Lai et al. 2020).

Contrary to that, Dealroom and Finch Capital (2019) show that the FinTech sector is more active in Europe than in Asia or the U.S., creating 150 billion euro in value. However, the regional reports demonstrate that the European market itself is also very diversified: FinTechs in smaller countries like Latvia, Lithuania, and Estonia are much more internationalized and diversified than those in larger countries (Laidroo et al. 2021).

This study is focused on the factors that support formation of new FinTech companies and the development of the whole FinTech sector in Poland. This is one of the biggest economies in the Central Europe, and the Polish market is also the biggest FinTech market in the Central and Eastern Europe, with an estimated value of 856 million euro (CEE Capital Market Leaders Forum 2019). It belongs to the group of post-communist countries which joined the European Union in 2004. When compared to the Western economies, it is characterized by higher speed of digitalization and a quicker adoption of financial innovations (GlobalData 2017). On the similar basis as the Asian tigers (Lai et al. 2020), the development of the FinTech sector seems to be not limited by the traditional perception of the financial system.

It is worth to note that the Polish FinTech market so far was mainly analysed by commercial analytic companies within technical reports (e.g., Flanders Investment and Trade (2018); Kliber et al. (2020); Microfinance Centre (2019)). To the best of the authors' knowledge, the only scientific publications about the Polish market include: Klimontowicz and Mitrega-Niestroj (2019), where the authors present the FinTech ecosystem, taxonomy of the Polish FinTech companies and the analysis of the financial market, Anielak (2019), who discussess innovativeness of Polish FinTechs, as well as Staszewska (2018) who analyses the interactions between the Polish FinTech and the banking sector. Thus the Polish FinTech sector has not been analysed thoroughly. Our article aims to fill this gap.

In the first step of our research, we provide an analysis of the publications that focus on the factors that support and accelerate FinTech start-ups' formation. Next, we analyse the Polish FinTech ecosystem concerning different criteria. Our analysis is based on the data obtained from the scientific databases and confronted with the results of the survey run among the companies from the Polish FinTech sector in January 2020 (see Kliber et al. (2020) for a detailed description of the survey results). Eventually, we identify the main obstacles that pose risk to the FinTech development. We analyse the statistical data and the responses from our survey to identify such threads in the Polish market. In the concluding section of our paper, we formulate policy implications.

We adopt descriptive and deductive approach in this study to address our main research goal, namely to identify obstacles and incentives for the FinTech growth in Poland. We apply a comparative analysis in order to assess the characteristics of the Polish market vs. other markets. We also apply a system analysis to identify the interactions between the economic, social, political and legal factors which all together create the environment for the FinTechs to operate.

The main contribution of this paper is the creation of a methodological framework to assess FinTech sector growth potential. We chose parameters and provided measures which allow to approximate country performance. Based on this, we identified risks and opportunities for FinTech firms in Poland. Our implications can be applied by the policymakers as we provide a screenshot of political, social, economic, and legal conditions for the sector's further development and we systematise it in the context of other countries' performance. In future research, we plan to apply this method and provide a broader regional and World FinTech potential map.

It is not certain how the COVID-19 and the resulting recession will impact the Fin-Tech sector. Some authors note that FinTech start-ups may struggle with access to funding. Knight and Wojcik (2020) say, however, that the pandemic has weakened incumbent financial institutions and stimulated the openness to digital finance solutions among consumers. Therefore, it is of the special importance to analyse the current situation of the sector and recognize its strengths and possible sources of risk in the economic and legal environment. These need to be improved to allow for the further development of this sector. Throughout the article, we try to defend the thesis that although Poland has a very good initial condition to support the formation of FinTech companies, there are obstacles that may restrain the development of the sector.

## 2. Literature Review

The literature focused on the development of FinTechs has been rising over a few years. First of all, the researchers are interested in the FinTech business models (Eickhoff et al. 2017; Laidroo et al. 2021; Lee and Shin 2018; Sannino et al. 2020). Liu et al. (2020) provide an overview of the 10-years history of research on the FinTechs topic, through the analysis of the 629 FinTech business model papers in the Web of Science database. The authors conclude that the most hot-topic in FinTech research are mobile payment, microfinance, peer-to-peer lending platform, and crowdfunding. They also suggest that the Blockchain and crowdfunding would dominate FinTech research in the nearest future.

There is a strand in the literature that discusses and analyses the interaction of Fin-Techs with banks and financial systems. FinTechs with the whole financial technology ecosystem have a disruptive impact on the financial services industry (Palmié et al. 2020). Bunea et al. (2016) analysed explicit mentions of competition from FinTech in the annual reports of the U.S. banks and found that there were no such remarks before 2016. The authors identified 14 banks that acknowledge being threatened by FinTech companies. The banks represented 3% of the banking sector by count but nearly a third of its assets. The results of Siek and Sutanto (2019) based on the quantitative analysis show that banks indeed have been disrupted by FinTechs since the emergence of such companies. Fintech were featured by superior value propositions and the concentration on customer satisfaction.

Another quantitative analysis of interrelationships between FinTechs and traditional institutions shows that finance services provided by the Internet tend to spill over first to the banking industry, then to the insurance industry, and finally to the securities industry (Chen et al. 2020).

In his paper, Anagnostopoulos (2018) argues that the competition between banks and FinTech has evolved into direct collaboration. Banks own legacy, financial expertise, infrastructure, and stable 'old' customer base, while FinTechs own agility, innovation, and future customer base. From the viewpoint of the banking sector, such an obstacle is the lack of clear regulations on IT security, while from the FinTechs side—the differences in culture and operational processes.

To summarize, the research shows that the traditional and new-finance co-exist and cooperate one with another, instead of competing (Bömer and Maxin 2018; Bunea et al. 2016; Siek and Sutanto 2019).

Eventually, there are papers in which authors discuss the factors that stimulate the growth of the FinTech sector, as well as the obstacles that prevent it. As this strand of the literature is crucial for our study, we present it in details in the following subsections.

### 2.1. Factors that Accelerate FinTech Formation and Development

We base our analysis on the results of the studies focused on the determinants of the FinTech formation. First, we refer to the study of Haddad and Hornuf (2018), which investigated the economic and technological determinants inducing entrepreneurs to establish FinTechs. Authors analysed a panel dataset that consisted of 1177 observations from 2005 to 2015 and covered 107 countries (including 26 FinTech from Poland, which gave the country the rang no 30). The authors proved that well-developed capital market, already available technical base and supporting infrastructure is crucial for FinTechs formation. On the other hand, a more fragile financial sector is also an incentive, since FinTechs and traditional financial services might act complementary in some market segments, such as a high-risk market loans. Other factors enhancing the FinTech formation were the favorable regulations and a larger labor market.

A similar study was performed by Laidroo and Avarmaa (2019). The authors determined location-specific factors associated with FinTech establishment intensity over the period 2007–2012 using Porter's diamond framework. They confirmed that FinTech formation intensity is greater in countries with stronger financial system and already available technology. Moreover, the formation intensity tends to be higher in smaller than in big countries. Other important incentives mentioned by the authors were: high tertiary education rate, university-industry cooperation, overall ICT readiness, and greater financial development level. The authors comment that although lower financial development may support FinTechs development in some areas, the existing infrastructure is necessary for the wider development of their services. The authors note also the importance of the more developed legal environment for the FinTech formation rate.

Eventually, in the most recent study, Cojoianu et al. (2020) investigated the influence of the new regional knowledge creation in both the IT and financial services sectors on the development of the FinTech sector, taking additionally into account the lack of trust in financial services incumbents. They analysed 21 countries and 226 OECD regions from 2007 to 2014. The authors confirmed that the new knowledge created both in the incumbent IT sector, as well as in the financial services sector, supports the FinTech emergence, but the importance of each source changes together with the growth of the FinTech sector. On the other hand, the authors found no statistically significant relationship between the FinTech emergence and the level of trust in financial services incumbents.

In Table 1, we display the list of the factors that support FinTech formation—according to the aforementioned studies. We present the factors tested, the variables used to approximate the influence of these factors, and the relationship found (where "+" denotes positive, "−" negative, while "0" no relationship at all).

**Table 1.** Review of the factors that support FinTech formation.

| Study | Factors Tested | Proxy Used | Relationship |
|---|---|---|---|
| (Haddad and Hornuf 2018) | Well developed financial market | GDP per capita, VCfinancing | + |
| | Available technology | Mobile telephone subscription, secure Internet services | + |
| | Fragile financial sector | Ease of access to loans | − |
| | Regulation | Regulation indicator from Fraser Institute database (variable taking value from 0 to 10, where higher values denote more market freedom) and strength of legal rights indicator from World Bank Doing Business database (variable taking value from 0 to 12, where higher value denote higher protection of borrowers' and lenders' right by collateral and bankruptcy laws) | + |

**Table 1.** *Cont.*

| Study | Factors Tested | Proxy Used | Relationship |
|---|---|---|---|
| (Laidroo and Avarmaa 2019) | Strong financial and ICT (information and communication) services clusters | Dummy variable indicating whether the mean ranking of the country in the list of financial centres was below or higher than 10; ICT service exports as % of service exports. | + |
| | Strong home demand | Percentage of citizens at the age of 15 years or older, who accessed their bank account via mobile phone or the internet; domestic market size index. | − |
| | Developed factor conditions | Tertiary education enrolment rates, university-industry cooperation, fixed-line availability, and overall ICT readiness | + |
| | Crisis (trust in traditional financial services) | Dummy variable equal to 1 if the country experienced a banking crisis over 2007–2017 | + |
| | Financial development levels | Financial freedom index (Heritage Foundation), Combined indicators of access to financing (based on GCI), Financial development index (IMF), Financial institutions index (IMF), Financial markets index (IMF), Stringency of capital requirements (World Bank), Supervisory power of regulatory authorities (World Bank), Banking activity restrictions (World Bank), Legal rights index (World Bank) | + |
| (Cojoianu et al. 2020) | Knowledge in IT sector | Sectional regional patent application counts in the IT sector | + |
| | Knowledge in financial sector | Fractional count of patent applications of asset managers, banks, insurance companies and stock exchanges | + |
| | IT sector productivity | The ratio of the gross value added (GVA) to total employment within IT sector | + |
| | Financial sector productivity | The ratio of the gross value added (GVA) to total employment within financial sector | + |
| | Level of trust in financial incumbents | The percentage of people answering "No" to the question in the Gallup Annual Survey: "In this country, do you trust financial institutions or banks? | 0 |

Note: Value: "+" in Relationship column denotes positive relationship, value: "−"—the negative one, while 0—lack of statistically significant relationship.

## 2.2. Obstacles and Risk Factors to FinTech Formation and Development

Based on the review of the literature presented in the previous subsection, we can identify several obstacles to FinTech development. The most common are: a lack of clear regulations, a lack of technical base, limitations considering skilled workers, and a mistrust in financial innovation (translating into the demand problem).

The problem of regulation and trust in financial innovations are linked one to another. When it comes to regulation the fact is that after the global financial crisis in 2008, the policymakers had been focused on the safety in finance (Zavolokina et al. 2016). This increased the transparency, data visibility, and trust of the customers. As a result, the clients prefer to locate their wealth with more trustworthy banks, and the latter have access to a much larger group of customers than the FinTechs. As noted by Zetzsche et al. (2017) and Hansen (2012), the first and foremost asset of financial services providers is their clients' trust. On the other hand, FinTechs themselves create many challenges for regulators. The new regulations should increase the trust of the customers and also should support the development of FinTechs. The process of creating regulations is complex and demands cooperation between the legislators and the FinTechs.

The other problem is the technical base. As already noted, the lower stage of financial development may accelerate FinTechs' expansion. However, as the sector expands, the infrastructure base supports the wide and full development of FinTech services.

Another crucial source of risk is the availability of qualified staff. The new knowledge created in the incumbent IT sector and the financial services sector is beneficial for the FinTech formation (Cojoianu et al. 2020). The knowledge creation depends on the qualified staff. For instance, Brown et al. (2019) analyse the impact of Brexit on small and medium enterprises, including FinTechs, and note the danger of the possible shortage of highly-skilled workforce, which is crucial for FinTech development. In line with this need, Sung et al. (2019) explores specifically the availability and opportunities for Fintech education and retraining in the UK.

## 3. Data and Research Methods

In our research, we apply both inductive and deductive methods, together with comparative and system analysis. The theoretical analysis of the factors that support FinTech growth and formation was based on the literature review including scientific articles, technical papers and press releases. In this part of the article we applied mostly the inductive method and comparative analysis.

In the empirical part of the paper we analyse the external quantitative data gathered during empirical research conducted by the World Bank, UNESCO, Statistics Poland, as well as through Gallup questionnaire. Yet, the data are confronted with the internal ones obtained through the survey conducted among Polish FinTechs in January 2020. The survey was run by Quantify in cooperation with QuantFin Foundation. We obtained responses from 48 companies. This part of analysis is based on deductive methods, namely descriptive statistical methods and comparative analyses. The calculations were performed using R.

Various definitions of FinTech can be found in the literature. Our definition of FinTech is consistent with the one used in Tirmaste et al. (2019) and Rupeika-Apoga et al. (2020), and is as follows: companies that provide financial services and have a clear, and generally innovative, information technology component in their business model. In consequence, our sample includes not only the startups but also some institutions of longer history. Such definition was used in order to obtain the consistent results with our partners from Estonia, Latvia, Lithuania and Russia, since the survey was run in cooperation. Hence, all the questions were structured in such a way that the results are comparable across the

countries.[2] The questions were based on the questionnaire used in Ankenbrand et al. (2018) and refered to the business model canvas of Osterwalder and Pigneur (2010). The model includes nine blocks representing the important parts of any business, i.e., key partners, key activities, key resources, value proposition, customer relationships, channels, customer segments, revenue streams, and cost structure (Osterwalder and Pigneur 2010). The questions in the survey addressed the first eight ones. Apart from that, we included sentiment questions, which are crucial for this paper and allowed us to confront the values of the indicators obtained from external databases with the opinion of the FinTechs. We asked companies to assess the challenges for their business on a scale from 1 (not pressing) to 10 (extremely pressing). The challenges comprised: finding customers, access to finance, costs of production or labour, availability of skilled staff, regulation, and expansion to international markets. The questions were based on the European Central Bank (2018) survey. The last part of the survey tackled the problem of the relationships of FinTechs with banks, as well as their projections on the future interrelationships. The proposed scenarios were taken from Basel Committee on Banking Supervision (2018).

The companies have filled the questionnaire online or via telephone interviews. The respondents of the survey were the managers or decision-makers. Most of the questions were closed-one, yet, in some of them, we asked for the opinion. Interested readers can find a technical description of the survey and its results in Kliber et al. (2020).

The adopted research method has the following advantages. First of all, we find in the literature the respective factors that have been identified by independent research groups based on quantitative analysis performed on different set of data and countries. Thus, we can suppose that their results are fairly robust. Next, we use the data from external databases to verify the position of Poland with respect to the identified factors. Based on the values of the indicators, we divide them into the ones that support or hamper the startup formation and sector development. Lastly, we confront the findings with the internal data obtained directly from the companies representing the sector. What is interesting, in some cases the sentiment of the respondents contradicts the external data. Therefore, we can say that the assessment of the opportunities and threads is a complex task and that the external data may not be sufficient enough to obtain a complete view of the internal situation of the sector.

## 4. Overview of the FinTech Sector in Poland

Poland is the biggest FinTech market in Central and Eastern Europe, with an estimated value of 856 million Euro (CEE Capital Market Leaders Forum 2019). The capital of Poland, Warsaw, is also a financial technology hub in the region and home to nearly 45% of startups in the country. Klimontowicz and Mitrega-Niestroj (2019), who divide Polish FinTech sector into three groups of: banks (and their FinTech accelerators), interbank and non-banks' entities (after Widawski and Brakoniecki (2016)) state that the non-banking FinTech sector alone reached net profit from EUR 10.5 to 14 million in 2017. The Polish FinTechs serve both individual customers and enterprises. Among the enterprises, the most important are financial institutions and small and medium-size companies (SME) (Klimontowicz and Mitrega-Niestroj 2019).

To identify the field of activity of Polish FinTechs we used the following classification.[3] We divide the FinTechs into eight groups, i.e.,

- Payment;
- Analytics;
- Banking infrastructure;

---

[2] The results of the analogous surveys run for Latvia and Estonia can be found respectively in (Rupeika-Apoga et al. 2020; Tirmaste et al. 2019) (our survey is the modification of the surveys presented in the reports for the Polish market), while the comparison of the FinTechs in the CEE in Laidroo et al. (2021).

[3] This classification is similar to that used in IFZ FinTech Study 2018 (Ankenbrand et al. 2018). Our classification is also compatible with the one used in FinTech Report Estonia (Tirmaste et al. 2019) and FinTech Study Latvia (Rupeika-Apoga et al. 2020), and the same as the one applied in (Kliber et al. 2020).

- Distributed ledger technology;
- Deposit and lending;
- Investment management;
- InsureTech;
- Accountech.

The payment group encompasses the companies dealing with mobile and online payments, mobile transfers and other form of payments. In the field of analytics we include enterprises that deal with data and business analytics including big data, machine learning, artificial inteligence used for automated advice, as well as chatbots. The companies from banking infrastructure are all those software companies in financial sector that prepare the user interface, enhance processing and produce infrastructure technology. Distributed ledger sector comprises cryptocurrency and blockchain technologies. In the group of deposit and lending we include crowdinvesting, crowdlending and invoice trading. According to our criteria, investment management sub-sector deals with robo-advising, social trading, hybrid models and advice-supported digital investing. InsureTech are companies that apply softwware technologies in insurance, while Accountech are the same for accounting.

At the beginning of 2020, we identified 233 FinTech companies in Poland. The source of our data were: Crunchbase and Cashless databases and the expert knowledge of our business partners: Quantfin and Quantify. Most of the FinTechs belonged to the payments sector (28.3%). The deposit and lending field was the second largest (22.7%). 17.6% of the enterprises were associated with banking infrastructure. Investment management and analytics were of almost the same size: 9.9% and 9.4%, respectively. The distributed ledger technologies subsector was relatively small, encompassing 5.2% of firms, while the smallest number of companies were classified as InsureTech and Accountech (3.4% each group).

In our study, we dealt with a subsample of the whole population (i.e., 233 companies). This subsample covered 48 companies that agreed to take part in our survey (the list of the participants can be found in the Appendix A). It reflected the distribution of the companies across the identified fields quite well: three main groups comprised companies dealing with payment (33.3%), deposit and lending (20.8%), and banking infrastructure (18.8%). The share of analytic companies was slightly larger than in the whole population (10.4% vs. 9.4%). The investment management sector was slightly unrepresented (6.3% as compared to 9.5% in the population), while the Accountech—overrepresented (6.3% vs. 3.4%). InsureTech and distributed ledger sector were the smallest groups (2.1% each—which means that we had only one respondent from each of the groups). In Table 2, we present the comparison of the companies in the whole population versus the one included in the sample. We run the chi-square Pearson's test and obtained $p$-value exceeding 0.8. Therefore we can suppose that the sample represents the population quite well, at least when the distribution across the FinTech types is considered.

**Table 2.** Comparison of the sample and population data.

|  | Population | Sample |
|---|---|---|
| Analytics | 9.4% | 10.4% |
| Investment management | 9.9% | 6.3% |
| Payment | 28.3% | 33.3% |
| Deposit and lending | 22.7% | 20.8% |
| Banking infrastructure | 17.6% | 18.8% |
| Distributed ledger technology | 5.2% | 2.1% |
| Accountech | 3.4% | 6.3% |
| InsureTech | 3.4% | 2.1% |

Note: To test whether the sample is representative, we run the chi-square Pearson's test. The $p$-value exceeded 0.8.

The distribution of the sample by the activity type is presented in Figure 1.[4] We emphasize that these companies operate in more than one sector. The classification relies heavily on the information found on the companies' websites or in other FinTech reports. Moreover, some companies did not indicate any of the pre-defined classes and chose to classify themselves using their classification.

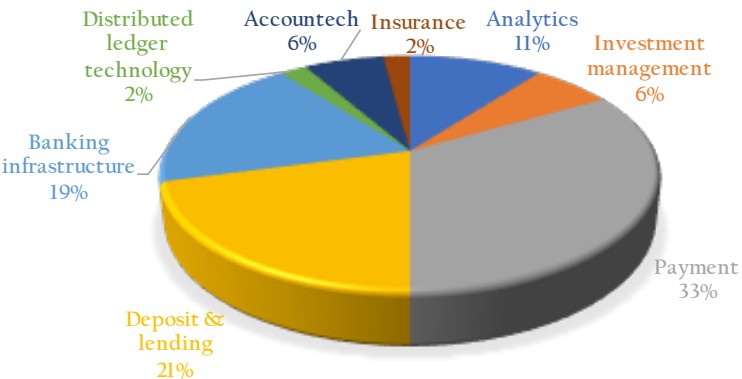

**Figure 1.** Field of activity of companies which completed the survey.

The FinTech sector in Poland is still a relatively young one—15% of our respondents were still under development (i.e., in a testing phase). The majority of our respondents were present on the market for 5 to 10 years (44% of the firms) or for 1 to 5 years (35%). The "mature" companies running their business from 10 to 15 years represent the lowest share (6%) in the group (see Figure 2, left panel).

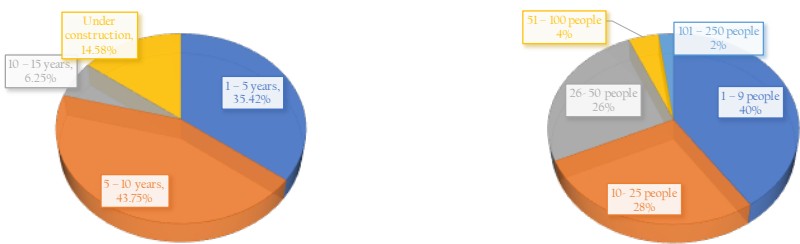

**Figure 2.** Maturity and size of FinTech companies.

In Figure 3, we present the maturity of companies across their fields of activity. We find that FinTechs which have been existing on the market for the longest time, are from the payment and analytic sectors. However, there are no analytical companies, as well as no banking-infrastructure ones in the under construction group. The payment companies are present in each maturity set. Accountech, investment management, and InsureTech companies belong to the group of the youngest ones (either under construction or up to 5 years on the market).

At the moment of writing this article, the FinTech sector in Poland has been dominated by small companies: enterprises of 1 to 9 employees constituted almost 40% of the sample (see Figure 2 right panel and Table 3). FinTechs that employ 10 to 25 people constitute 27.1% of the sample, and the ones that engage from 26 to 50 people—25%.

Table 3 presents the size of the FinTechs across their activity. The smallest companies (1 to 9 employees) were most often present in payment, deposit and L lending, insurance, and investment management. The banking infrastructure sector was represented by larger firms (from 10 to 25 employees). In analytics, enterprises employing from 26 to

---

4　The survey allowed the respondents to choose more than one option as an area of activity.

50 people dominated. The largest companies belonged only to banking infrastructure or payment groups.

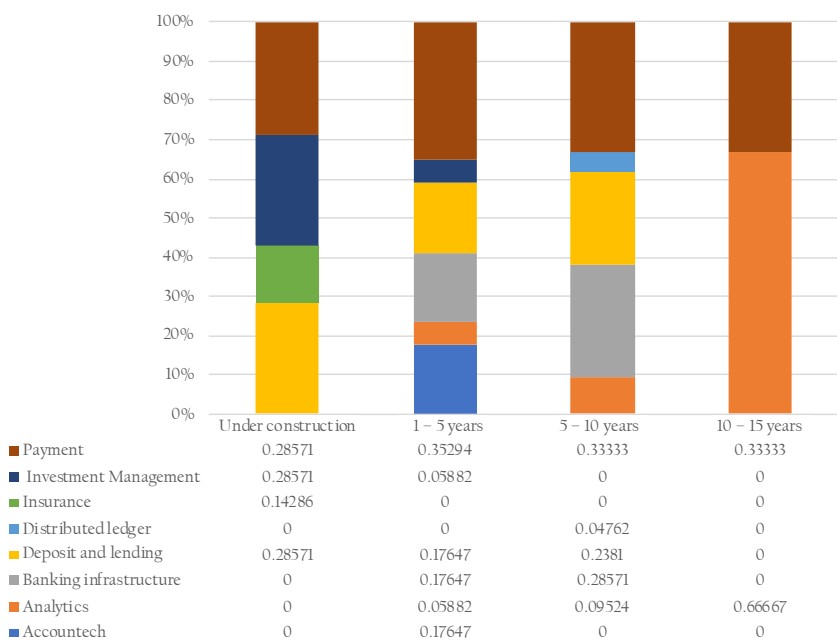

| | Under construction | 1 – 5 years | 5 – 10 years | 10 – 15 years |
|---|---|---|---|---|
| ■ Payment | 0.28571 | 0.35294 | 0.33333 | 0.33333 |
| ■ Investment Management | 0.28571 | 0.05882 | 0 | 0 |
| ■ Insurance | 0.14286 | 0 | 0 | 0 |
| ■ Distributed ledger | 0 | 0 | 0.04762 | 0 |
| ■ Deposit and lending | 0.28571 | 0.17647 | 0.2381 | 0 |
| ■ Banking infrastructure | 0 | 0.17647 | 0.28571 | 0 |
| ■ Analytics | 0 | 0.05882 | 0.09524 | 0.66667 |
| ■ Accountech | 0 | 0.17647 | 0 | 0 |

**Figure 3.** Maturity of companies by field of activity.

**Table 3.** Number of employees versus the field of activity.

| Field of Activity | Number of Employees | | | | | |
|---|---|---|---|---|---|---|
| | 1–9 | 10–25 | 26–50 | 51–100 | 101–250 | No Answer |
| Accountech | 2.1% | 4.2% | - | - | - | - |
| Analytics | 2.1% | 2.1% | 6.3% | - | - | - |
| Banking Infrastructure | 4.2% | 6.3% | 4.2% | 2.1% | 2.1% | - |
| Deposit and lending | 8.3% | 6.3% | 4.2% | - | - | 2.1% |
| Distributed ledger | - | - | 2.1% | - | - | - |
| Insurance | 2.1% | - | - | - | - | - |
| Investment Management | 4.2% | - | 2.1% | - | - | - |
| Payment | 16.7% | 8.3% | 6.3% | 2.1% | - | - |
| Total | 39.6% | 27.1% | 25.0%% | 4.2% | 2.1% | 2.1% |

Source: FinTechs in Poland: Insights, Trends and Perspectives: (Kliber et al. 2020).

The Polish FinTechs are not very internationalized. The employees of the majority (87.5%) were working in Poland. Only four companies indicated that a part of their workers operated from abroad (in these cases the ratio of workers from Poland was: 98%, 90%, 10%, and 2%).[5]

The trends in the workforce suggest that the FinTech sector in Poland is still expanding. Although most of our respondents (54.2%) did not note a change in the number of employees between 2018 and 2019, only 6.3% reported a moderate decmidrule. In the rest of the cases, the companies reported either moderate (22.9%) or large (16.7%) growth.

---

[5]    The lack of internalization is visible also when we analyse the country of registration of the FinTechs. 46 companies from our sample were registered in Poland, one in the Czech Republic, and one in Belgium. Most of the Polish FinTechs focused their business on the Polish market (77% of the respondents), and only 33% were oriented on international clients.

To sum up, the FinTech sector in Poland is still growing. The micro and small enterprises dominate the market. Start-ups constitute a substantial share of the business. The most mature companies are from the analytic group, while the largest deal with payment or banking infrastructure. Therefore, we find it important, to investigate whether Poland satisfies the condition to support further formation of the new FinTechs companies, which in turn will translate into the growth and development of the whole sector in the country.

## 5. Factors That Support FinTech Formation and Development

In this section, we identify triggers and opportunities for the FinTech sector development in Poland. The analysis is done with respect to the factors identified in the literature and specifier in Table 1. We discuss them and illustrate them with the statistical data from the external scientific databases and primary data from our survey.

In the group of factors that accelerate the FinTech growth, we include the stage of development of the financial system, the very secure banking system unwilling to give loans to risky enterprises but willing to cooperate with the FinTechs, education level (high tertiary education enrolment rate, high rate of technical students), as well as the available technology (knowledge in the IT sector) and the openness to financial innovation.

### 5.1. Stage of Development of the Financial System in Poland

One of the crucial factor which is supportive for FinTech formation and development is the adequate level of the financial system development. It is approximated inter alia by the number of commercial bank branches and bank accounts per 1,000,000 (see: Figure 4), by the GDP per capita, ease of access to loans, as well as indices of financial development.

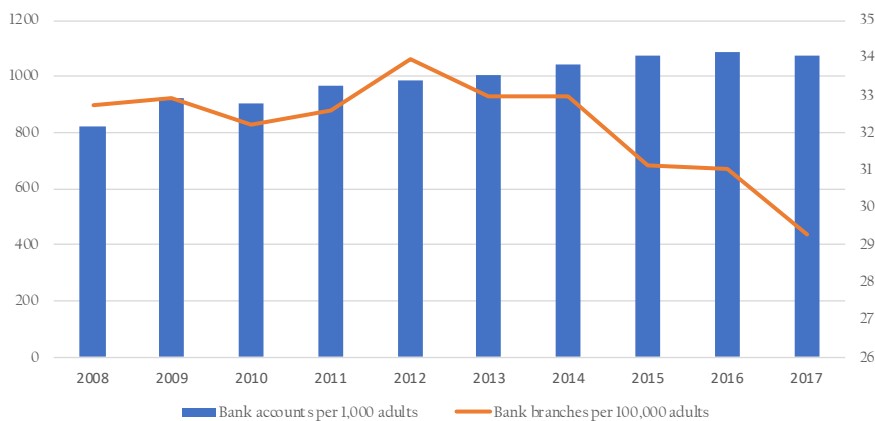

**Figure 4.** Number of bank accounts and bank branches per 1,000,000 in Poland over the period 2011–2017 (World Bank: Financial Development database)

The crucial feature of the financial system in Poland is the well-grounded banking system which follows the German approach to financing. The main source of financing of the companies is the bank credit. Bank transfers have been still the most-used payment method accounting for 50% of all payments in 2019 (JPMorgan 2019a). This is, inter alia, the consequence of legal solutions that require business payments to be executed via bank transfer.

However, there are some characteristics in the Polish financial market, in particular in terms of consumer behavior, which makes it somehow more promising to FinTech development. It seems that there are sharp differences between consumers' habits and approaches in Germany and Poland. The report of J.P. Morgan shows that German consumers are featured by the conservative approach to spending, and Germany's e-commerce sector is highly influenced by habits formed in the pre-internet era (JPMorgan 2019b). On contrary, Polish consumers are one of Europe's early adopters, eager to migrate to digital-only payment options and to launch the contactless technology (GlobalData 2017). Mobile payment

systems have been relatively popular. Both card payments and digital wallets had been growing in popularity fast, with forecasts of 25% and 33% of the compound annual growth rate, respectively (JPMorgan 2019a). These tendencies have even strengthened during the outbreak.

Table 4 shows trends in e-commerce in various European countries. Within the group, Poland has a rather moderate position. E-commerce market value and mobile commerce market size are relatively low when compared to other European countries. Mobile commerce has only 11% share of overall e-commerce sales. The internet access is second lowest within the group of countries included in the reports. The popularity of smartphones is a little below the average. The number of cards per person is comparable to Italy but higher than in France or Germany. The predicted growth rate of the e-commerce segment is two-digit. All these features favor strong market potential to develop. Bank transfers which are Poland's most-used payment method might be easily proxied by e-payments soon.

**Table 4.** E-commerce payments trends in European countries.

|  | E-Comm | M-Comm | Internet | SPHN | Cards | GR |
|---|---|---|---|---|---|---|
| Austria | 9.1 | 0.7 | 88 | 65.0 | 1.51 | 8 |
| Belgium | 10 | 1.5 | 89 | 69.7 | 1.95 | 8.5 |
| Czechs | 4.4 | 2.4 | 85 | 67.3 | 1.11 | 16 |
| Denmark | 15.4 | 4.9 | 97 | 78.2 | 1.57 | 10.5 |
| Finland | 8.5 | 2.8 | 94 | 76.0 | 2.3 | 11 |
| France | 81.7 | 17.2 | 88 | 67.8 | 0.94 | 10.5 |
| Germany | 73 | 19.7 | 91 | 71.0 | 0.52 | 7.3 |
| Ireland | 7 | 2.9 | 83 | 71.0 | 1.37 | 8.6 |
| Italy | 21.2 | 7 | 73 | 68.5 | 1.09 | 14 |
| Luxembourg | 0.74 | 0.14 | 98 | 70.5 | 4.35 | 8 |
| Netherlands | 22.5 | 4.5 | 96 | 71.0 | 0.18 | 11 |
| Norway | 10.9 | 4.3 | 99 | 76.1 | 2.92 | 13 |
| Poland | 9.9 | 1.1 | 78 | 66.5 | 1.03 | 10 |
| Portugal | 4.3 | 0.43 | na | 68.0 | 2.66 | 12 |
| Spain | 30.3 | 11.5 | 87 | 69.5 | 1.64 | 13.5 |
| Sweden | 12 | 4.9 | 97 | 74.0 | 1.89 | 9 |
| Switzerland | 10.1 | 2.7 | 91 | 73.5 | 1.95 | 7.5 |
| UK | 178.5 | 91 | 95 | 70.8 | 2.48 | 9 |

Note: Table is based on J.P.Morgan's Payment reports (JPMorgan 2019c). E-comm denotes e-commerce market value, M-comm denotes mobile commerce market size (both in billions of euro), Internet denotes internet penetration (in %), SPHN is for smartphones' penetration (in %), Cards denotes number of cards per capita, and GR denotes the predicted e-commerce compound annual growth rate 2017–2021.

### 5.2. FinTechs and Banks

Together with the rapid development of financial innovations, a discussion about the possible threat that FinTechs pose to the established banking sector has emerged. However, as we pointed out in Section 2 of this article, the worldwide research shows that the traditional and new-finance co-exist and cooperate one with another, instead of competing (Bömer and Maxin 2018; Bunea et al. 2016; Siek and Sutanto 2019). Based on the analysis of 14 case-studies of FinTech-bank cooperation in Germany, Bömer and Maxin (2018) identified three main reasons why such collaboration is profitable for FinTechs. To begin with, it enables FinTechs to enter the market. Next, cooperation with a bank increases the FinTechs' profits. Finally, banks enable new FinTech products.

The situation in Poland resembles the one observed worldwide. The authors of the "FINTECH in Poland" market survey (Flanders Investment and Trade 2018) indicate that with respect to the banking sector, Poland is a regional leader in high-tech pioneering solutions. Figure 1 shows the activities of the Polish FinTechs, which operate in various areas (payment, deposit and lending, banking infrastructure, investment management). They are already prepared to support banks with customer relationships, offering better or

more personalized products. In the future Fintechs might be able to replace banks entirely. The results of another international survey on mobile banking conducted by ING in 2015 (ING 2015), show that when the number of users of mobile banking is considered, Poland is the third country in Europe; 60% of smartphone users in Poland had already used mobile banking or expected to use it. Only the Netherlands and the UK show higher engagement in mobile banking services (67% and 63%, respectively).

One of our survey question was How do you see that FinTechs change traditional banks? The answers show that FinTechs do not perceive themselves as competitors. They rather tend to collaborate with the traditional banks. The majority of respondents (73%) claim that traditional banks will unavoidably endorse new technologies, modernize, and digitalize their services. More than half (60%) claim that new business providing specialized services would emerge, 44% of respondents expect the role of traditional banks to be restricted to offer commoditized services only. In such scenario direct customer relationships will be handled by other entities (FinTechs). For a minority of respondents traditional banks would become either irrelevant (17%), or disappear (19%). Then new technology-driven firms will be created in their place.

When the cooperation with banks is considered only 17% of FinTechs' managers stated that they do not have any common interests with banks. Among those who collaborate, the majority create new IT solutions, offer and aggregate bank products. They sell analytical tools, mobile applications, or programs responding to the challenges of banks related office issues. Naturally, FinTechs themselves use bank products such as traditional bank accounts. Some also interact with banks indirectly through leasing companies or brokerage houses.

Our results corroborate the findings of Staszewska (2018), who noted that FinTechs are eager to cooperate with banks (and vice-versa). Obviously, the FinTechs cannot yet compete with well-settled banks when the convenience and security is taken into account. So far, as everywhere in the world the role of Polish FinTechs is to "disrupt" the financial sector and change relations among the market participants in the near future. So far most FinTechs in Poland collaborate with banks, either as customers or as supporters. They are linked through friendly cooperation and derive mutual benefits.

### 5.3. Trust in Financial Incumbents

Although Cojoianu et al. (2020) showed that the level of trust in financial incumbents is not a necessary factor supporting FinTech formation and development, other researchers claim that trust is the most important factor that may encourage or discourage clients to put their money into a financial institution (Hansen 2012; Zetzsche et al. 2017). In the case of Poland, as we already said, most of the FinTechs collaborate with banks. What is important, Polish citizens trust banks more than the citizens of Western European countries. Although in 2009 the decrease of the level of trust in banks was sharper in Poland than in the global market, already a year prior to the COVID pandemic, it was significantly higher than worldwide (Piotrowski 2020).

Taking the above into account, we can formulate conclusion that the high level of trust in banks is a factor supporting the formation of new FinTechs that plan to cooperate with banks, as well as the future development of the sector (especially the subsector classified as banking infrastructure).

### 5.4. Factor Conditions—Trends in the Education

When it comes to the availability of qualified staff, we observe a growing trend of tertiary educational attainment. Analyzing the data from the Ministry of Science and Higher Education, we can see that Computer Science, Economics as well as Finance and Accounting have been ranked among the most popular fields of study for over 10 years.

Table 5 presents data collected by the ministry on recruitment to state and private universities in the last 10 years. As it is easy to observe, that Computer Science has been an unquestionable leader for 9 years, being the first among the most popular fields of study. Economics also has a stable 4–6 positions, and Finance and Accounting ranks 6–10

depending on the year. As one can see in the Figure 5, the percentage share of all candidates for the three analyzed fields ranged from 12% to 18% of all accepted candidates It is worth noting that not the entire ranking looks so stable. For instance, Construction has dropped from the first place in 2011/2012 to the 9th in recent years, while Education has dropped out from the top 10 most popular fields of study.

**Table 5.** Selected results of recruitment for higher education in 2011–2021. Showing the place in the popularity ranking and the number of candidates for Computer Science, Economics, Finance and Accounting field of study

| Academic Year | Total No. of New Students | Computer Science Rank | Candidates No. for Computer Science | Economy Rank | Candidates No. for Economy | Finance and Accounting Rank | Candidates No. for Finance and Accounting |
|---|---|---|---|---|---|---|---|
| 2011/2012 | 555,439 | 3 | 29,888 | 6 | 21,523 | 7 | 13,610 |
| 2012/2013 | 549,443 | 1 | 30,639 | 6 | 20,202 | 8 | 14,729 |
| 2013/2014 | 476,809 | 1 | 31,782 | 6 | 17,298 | 6 | 16,138 |
| 2014/2015 | 462,681 | 1 | 30,309 | 4 | 16,061 | 7 | 15,535 |
| 2015/2016 | 446,012 | 1 | 35,137 | 6 | 15,649 | 6 | 15,512 |
| 2016/2017 | 436,316 | 1 | 38,285 | 7 | 15,459 | 8 | 14,873 |
| 2017/2018 | 429,114 | 1 | 42,434 | 4 | 17,938 | 7 | 15,014 |
| 2018/2019 | 416,153 | 1 | 42,759 | 4 | 18,773 | 10 | 16,275 |
| 2019/2020 | 424,328 | 1 | 32,680 | 6 | 17,143 | 9 | 17,642 |
| 2020/2021 | 428,609 | 1 | 33,687 | 6 | 16,708 | 7 | 19,998 |

Source: Statistics Poland databases.

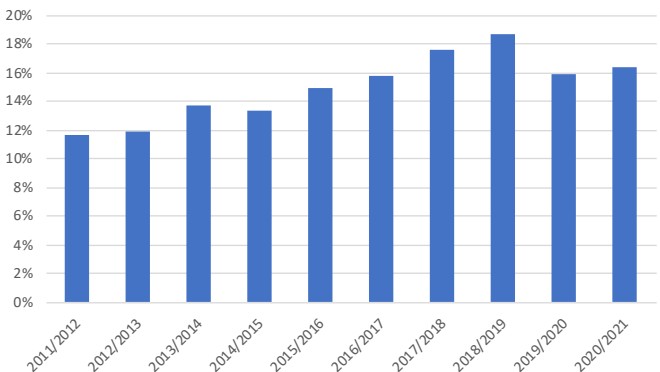

**Figure 5.** Number of candidates for Computer Science, Economics and Finance and Accounting as a percentage of all new admissions to higher education.

The constant and very high interest in these fields of science should promise a large number of qualified employees for the FinTech industry. However, as we show in Section 6, one of the high-risk factors identified by our respondents is the availability of the skilled workers. We discuss the possible reason for this discrepancy in Section 6.2.

### 5.5. Available Technology and the Openness to Financial Innovation

Another very important factor supporting the FinTech formation is available technology (Cojoianu et al. 2020; Haddad and Hornuf 2018; Laidroo and Avarmaa 2019). It is usually approximated with the secure Internet services or fixed-line availability. In Figure 6, we present the enormous growth of the number of secure Internet servers in Poland over the period 2011–2019. We note, that the cloud servers (e.g., Azure, AWS) are located outside of Poland, which can be a factor of data storage cost, yet we assume that this is not an obstacle per se. Thus, we can assume that the technology already available in Poland is a factor contributing to the FinTech sector development.

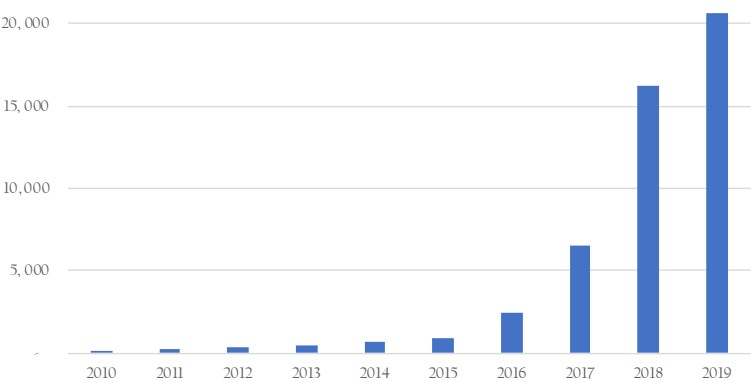

**Figure 6.** Secure Internet server in Poland (World Bank database).

*5.6. Triggers Behind the Sector Development Identified by the FinTechs*

In the previous subsections, we outlined the opportunities for FinTech formation, referring to the factors identified in the literature. In our survey we also asked the companies to identify the factors that can be considered triggers behind the sector development. Figure 7 presents the answers given by our respondents. The most frequently chosen ones were digitalization of financial services, expansion of FinTech beyond traditional financial services and rising number of payment options at retailers.

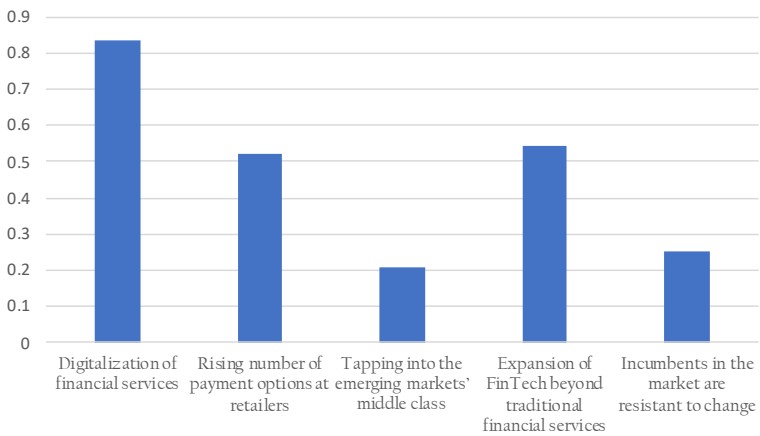

**Figure 7.** Triggers behind FinTech development.

Additional answers provided by our respondents were:

- Innovation;
- Regulations favoring open banking;
- Customer-orientation (FinTechs put emphasis on user experience and quality improvement);
- Qualification (FinTechs gather highly qualified staff).

Figure 8 shows the distribution of answers by maturity (already running versus under construction). Since the sample of the companies under construction is much smaller, we express the answers in the form of percentage—i.e., what share of the group chose the given option (each could have been chosen more than once). Thus, we are able to compare the opinions of the companies who already operate on the market and those, who are planning to enter it. We note some discrepancies: the companies that already operate chose digitalization of financial services and rising number of payment options at retailers most frequently, while the new companies see their strength in the fact that incumbents in the market are resistant to change.

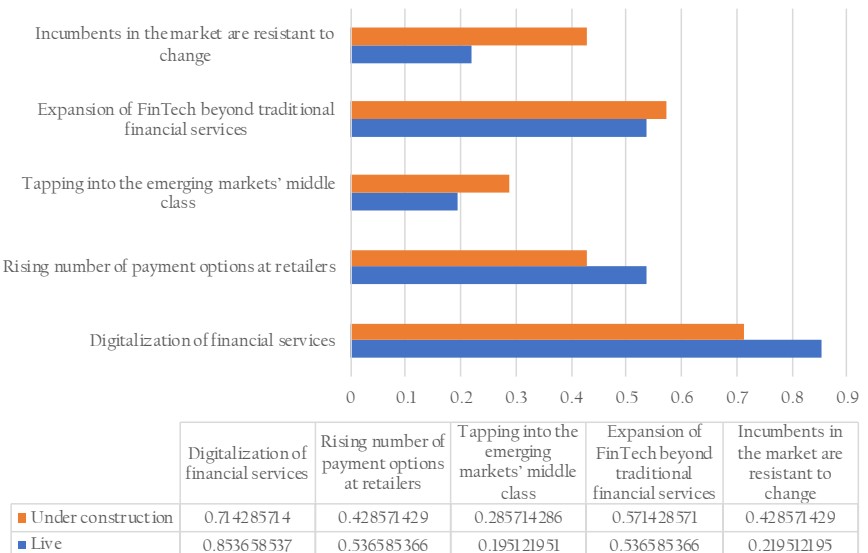

| | Digitalization of financial services | Rising number of payment options at retailers | Tapping into the emerging markets' middle class | Expansion of FinTech beyond traditional financial services | Incumbents in the market are resistant to change |
|---|---|---|---|---|---|
| Under construction | 0.714285714 | 0.428571429 | 0.285714286 | 0.571428571 | 0.428571429 |
| Live | 0.853658537 | 0.536585366 | 0.195121951 | 0.536585366 | 0.219512195 |

**Figure 8.** Triggers behind FinTech development—distribution of answers across the maturity of the companies.

Thus, we can conclude that the companies themselves see the opportunities for their development in the future development of the financial system, the openness of customers to innovation, but also the reluctance of the incumbents to be innovative themselves. Therefore, the FinTech can fill up the demand from the customers' side.

*5.7. Positive Trends in Legislation*

Some credit, in terms of financial openness, should be given to institutional efforts. A government institution that plays the role of the financial market supervisor and takes part in the respective legislative works is The Polish Financial Supervision Authority (Urząd Komisji Nadzoru Finansowego, further: the KNF). Its two main initiatives are Special Task Force for Financial Innovation in Poland, and Innovation Hub. The goal of the first one is to identify obstacles in the institutional framework. The second one (the Hub) provides institutional support for companies from the FinTech sector. It helps, among others, with interpreting the regulations, obtaining licenses, and maintaining adequate client protection. All FinTech start-ups who plan to introduce their innovative products into the financial market under the KNF supervision and seek institutional support can qualify for the Innovation Hub Programme.

**6. Challenges for the FinTechs' Growth in Poland**

When it comes to the challenges and risk factors that can possibly hamper FinTech formation and sector development in Poland, the most important ones are: regulations, possible home demand problems, factor conditions (problems with finding qualified stuff, which translates into the education and university-industry cooperation—see Table 1), and problems with financing.

One of the questions in our survey tackled the specific problems that the FinTech encounter (see Figure 9 and Table 6). The most pressing ones appeared to be the home demand problem, i.e., finding customers. The companies ranked the risk of finding skilled staff and inadequate regulations equally high, but slightly lower than the customer-finding. As a high-risk factor, the FinTechs pointed also increasing production costs and expansion to international markets (which can be associated with home demand problems). The enterprises worried less about access to finance and competition. In the open-answer question, they also enumerated the high cost of data (8 points on the 1 to 10 scale), problems with the large organizations' attitude towards FinTechs (10 points), and once again—the regulations (9 points).

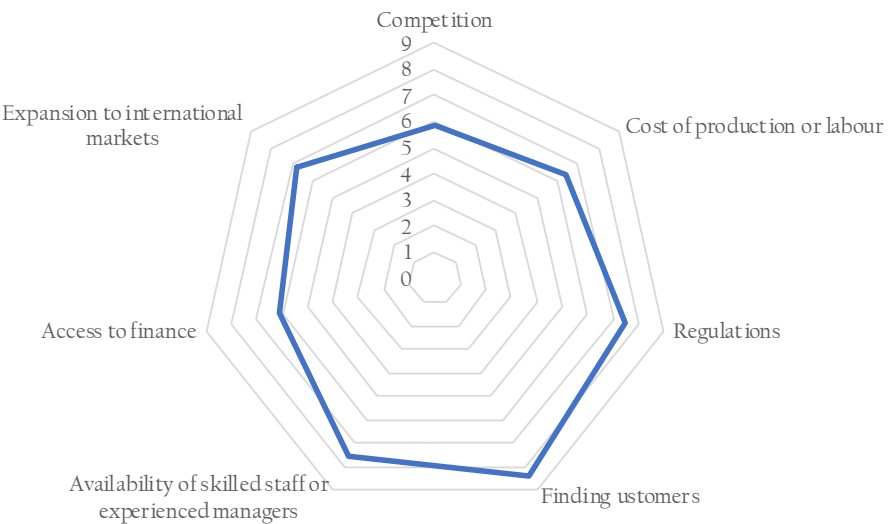

**Figure 9.** Specific problems and their pressure (means).

**Table 6.** Specific problems and their pressure—by FinTech type.

|  | **Accountech** | **Analytics** | **Banking Infr.** | **Dep. and Lend.** | **Inv. Manag.** | **Payment** |
|---|---|---|---|---|---|---|
| Competition | 4.67 | 5.00 | 6.00 | 7.10 | 4.67 | 5.81 |
| Finding customers | 9.67 | 8.60 | 7.44 | 9.00 | 7.00 | 8.56 |
| Access to finance | 6.00 | 5.40 | 6.11 | 7.50 | 7.33 | 5.13 |
| Cost of production/labour | 6.67 | 6.40 | 7.11 | 6.30 | 4.67 | 6.88 |
| Availability of staff | 7.00 | 8.60 | 7.89 | 7.60 | 7.00 | 7.38 |
| Regulation | 4.67 | 8.00 | 8.11 | 8.30 | 8.33 | 7.25 |
| Expansion to international markets | 7.67 | 7.20 | 7.56 | 7.00 | 6.33 | 6.38 |

Source: FinTechs in Poland: Insights, Trends and Perspectives: (Kliber et al. 2020).

We further compared the answers across the six largest FinTech types to check whether the problems were valued equally by each group (Table 6). There are some differences in the answers. Accountech, payment, and deposit and lending firms ranked the risk of finding customers the highest, as compared to other FinTechs. Accountechs were not concerned about the regulations and competition as much as other groups. The competition was also ranked as a relatively low risk-factor by investment and management. The whole sector, however, was approximately equally worried about the possibility to find competent staff.

### 6.1. Home Demand Problems

As already stated, the respondents of our survey pointed out home demand as one of the most pressing ones. However, we note that the companies from investment management as well as banking infrastructure rated it lower than e.g., the Accountech or Deposit and Lending ones.

In Figure 10, we display the change in mobile phone usage for payment and money transfer between the 2014 and 2017. The number of people using mobile to pay bills increased almost four times, while those using mobile to send money—almost twice.

In Section 5.1, we mentioned also that Polish consumers are one of the Europe's early adopters of financial innovations. It is worth to stress the early (pre-pandemic) success of BLIK payments. BLIK is a Polish FinTech that uses the domestic automatic clearing houses (ACH) to enable instant payments and mobile transfers. The initiative was launched in 2015 as a joint venture of the six largest Polish banks, while at the moment of writing this paper, it covers all major banks and payment institutions in Poland. According to Baba et al. (2020), during the second quarter of 2020, BLIK executed over 1 million transactions per day and was available to 13.1 million registered users. Therefore, it seems that the home demand should not constitute a big thread for the FinTechs from the payment sector as well.

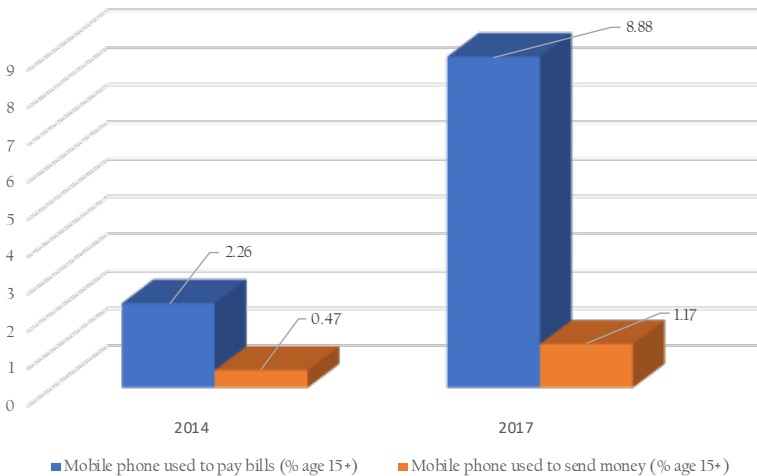

**Figure 10.** Mobile phone usage for payment and money transfer: 2014 vs. 2017 (source: World Bank Financial Development database).

*6.2. Knowledge and Talent*

Poland struggles with the issues related to innovation and talent retention. In terms of funding R&D Poland is among laggard Community economies, spending only half of the European Union average (see Table 7). At the same time, the dynamic index of technicians and researchers in R&D growth is a bit better for Poland (1.03 and 1.10) than for the Community on the whole (1.01 and 1.04); yet taking into account that Poland should be still catching up to the leaders, it seems quite moderate result to support innovative sectors development.

**Table 7.** Research and technology indicators.

| Year | Scientific and Technical Journal Articles | Technicians in R&D (Per Mln People) | R&D Expenditure (% of GDP) | Researchers in R&D (Per Mln People) |
| --- | --- | --- | --- | --- |
| 2011 | 25,735.0 | 359.8 | 0.7 | 1675.1 |
| 2012 | 27,969.6 | 420.6 | 0.9 | 1752.7 |
| 2013 | 30,026.1 | 384.7 | 0.9 | 1873.1 |
| 2014 | 31,773.3 | 438.5 | 0.9 | 2064.0 |
| 2015 | 33,116.4 | 443.6 | 1.0 | 2171.6 |
| 2016 | 34,838.7 | 399.8 | 1.0 | 2320.8 |
| 2017 | 34,675.7 | 415.3 | 1.0 | 3019.1 |
| 2018 | 35,662.6 | NA | 1.2 | 3106.1 |

Source: Scientific and technical journal articles: National Science Foundation, the rest of indicators: UNESCO.

Based in the Global Competitiveness Report, Poland is ranked 59th World economy in terms of Innovation and sophistication factors, placing it below European and North American regional average. In particular, Poland is ranked poorly for the university-industry collaboration in R&D and government procurement of advanced technology products. Slight trend of improvement can be noticed for the innovation capacity, company soundings on R&D, and PCT patents applications. Moreover, the fact that Poland is especially badly rated for both keeping (rank 89) and attracting talents (rank 113) appears to by an obstacle for any innovative sector to flourish.

According to Global Entrepreneurship Monitor data for 2019, the extent to which training in creating or managing SMEs is incorporated within the education and training system at primary and secondary levels is actually getting worse in Poland, compared to previous years (indices value 2.16 in 2019, 3.07 in 2013). However, it is getting better in higher education such as vocational, college, business schools, etc. (4.57 in 2019). Moreover, the extent to which social and cultural norms encourage or allow actions leading to new

business methods or activities that can potentially increase personal wealth and income are considered very low in Poland. This means that despite the relatively decent education system (see Section 5.4), the potential for sophisticated branches may be retained by lack of highly qualified stuff, institutional weakness, social values, and R&D policy.

### 6.3. IT and Financial Sector Productivity

Interestingly, the value of Financial Development Index by IMF for Poland is currently at the similar level as in the 1980s (0.47 in 2018), after a temporary deterioration in the 1990s and 2000s. This places Poland far behind the leaders such as Switzerland (0.96) or UK (0.90), but also below the European average value of the index, 0.52. The evaluation of Financial Institutions Index for Poland is higher than the value of Financial Markets Index (respectively 0.59 and 0.32).

The capacity of Polish firms to export service related to finance and ICT may approximate the conditions of the sectors and their productivity. According to WDI data, Poland improved its performance in terms of high technology exports (7% in 2011 to 11% in 2019) as a share of manufactured exports. As such it is closer to the European Union regional average of 16%. In terms of ICT goods, these remain consequently around 7% of total goods export since 2011, which is above the mean value in European Union (5% in 2018). However, a sneak peak of the WDI data on insurance and financial services exports indicates that these are only 2% in commercial services exports, which is less than 7% average in the UE. Thus, we can consider that both financial sector internal and export performance need improvement to better accommodate FinTechs.

### 6.4. Regulations

We note that most of the companies ranked the regulations as one of the highest risk factors. Yet, only 37.5% of them admitted that the current regulations restrict their activities. FinTechs criticized the imprecision of the regulations, denoted their ambiguity and the fact that they are often out-of-date (the regulations are backward, not taking into account rapidly changing reality—as one of the respondents wrote). The companies complained also about the unnecessary bureaucracy (e.g., the companies from the payment sector, who struggled to acquire the status of a payment institution). Some sectors were more affected by the lack of regulations—for instance, the leasing companies that were struggling for the regulation that would make it possible to sign leasing contracts online. Those enterprises that offer complementary services to banks, reported that the legislators very often neglect the existence of the FinTechs. In yet another question, FinTechs from the sector of analytics, banking infrastructure, deposit and lending, as well as payment admitted that they felt subject to relatively high compliance regimes, as compared to their competitors (for details see the report (Kliber et al. 2020)). All the respondents would appreciate it if the legislators consult them before implementing further modifications to the existing law.

In Table 1, we presented the proxies used in the literature, to describe the state of the regulations favouring FinTech formation and development. According to the Fraser Institute data for 2018 (data for 2019 and 2020 are not available at the moment of writing this report), the value of the Regulation indicator for Poland amounts to 7.32, while Legal system and property rights—to 5.99. The value of the first index is comparable to the one obtained by Hungary (7.42), but evidently lower than the scores of the Czech Republic, as well as small Baltic Republics (in each case higher than 8). Furthermore, when it comes to the Legal system and property rights—Poland stays behind the other CEE economies, for which the analogous variable exceeds 6 (Hungary and the Slovak and Czech Republics) or even 7 (Baltic republics). Yet another indicator used in the literature is legal rights from the World Bank database, which describes the degree of protection of the lender and borrower in the case of bankruptcy. The value of the index reached by Poland has not changed since 2014 and amounts to 7, which is the same value as the one reached by the Czech Republic, Slovakia and Estonia, higher than Lithuania (6), but lower than Latvia and Hungary (9).

The complexity and uncertainty of the regulatory environment hinder FinTech development in Poland (KNF 2018). This was confirmed by the respondents of our survey (one of them said that they will contribute to the ongoing dialogue on the introduction of legal regulations).

Polish authorities undertake legislative and regulatory activities to mitigate these risks. So far, Poland has implemented European FinTech directives on Payment Services Directive (PSD2), Anti-Money Laundering and Counter-Terrorist Financing (AML/CTF), and the Markets in Financial Instruments Directive (MiFID II). Other Community regulations, such as eIDAS[6] are directly applied in all Member states and require supervision. There are plans to introduce new legal measures for the FinTech sector, such as regulatory sandbox, artificial intelligence, and distributed processing technologies in the supervision of the financial sector (KNF 2020b).

Postulates for regulatory agenda include decreasing formalities, limiting over regulation, and clarifying the rules for new entities. It is required to recognize and secure non-banking FinTech entities' presence on the market by enabling them to access basic needs such as maintaining bank accounts (for instance, sometimes banks refuse to open bank accounts for cryptocurrency trading platform organizers).

Entrepreneurs in Poland face the problems of lengthy procedures and stale written-form requirements. Lack of new regulations related to registering accessibility, data processing, or digital identity implies the use of outdated legal solutions (KNF 2020b). For instance, business information offices should be allowed to outsource using modern technological solutions and to enable creditors to send payment requests electronically in all cases (email, SMS, MMS). Moreover, there is a need to create an institutional framework to prevent identity theft, which hampers FinTech industry development (KNF 2020a).

The KNF Working group identifies numerous needed changes in the legal environment of FinTech (Kliber et al. 2020). We may divide them into new rules, amendments, and a need for interpretation. Some issues require to be regulated, e.g., new instruments for start-up funding, the publication of interpretation and review of rules for the KNF, or legal advice for new trans-border PSP. The amendments to existing rules, according to the KNF Working group, should cover, for instance, the substitution of written form with a digital form of documents, the taxation of crypto-assets, or tokenization of bills, cheques, or bills of exchange. Eventually, practice change is required in areas related to the length of getting a license for payment service processing, robo-advice, creation of new securities, etc. In reference to the answers from the survey, we can conclude that not only the FinTechs see the need for future regulations change.

### 6.5. Access to Capital

Haddad and Hornuf (2018) show that access to financing is a crucial factor that encourages the FinTech formation and used it as a proxy for the financial market development. As noted in Lai et al. (2020), well-functioning capital and venture capital markets are crucial to entrepreneurship, the high-tech industry as well as innovation. For these reasons we decided to describe this aspect of the Polish market.

First of all, venture capital (VC) market of Poland is perceived as "inexperienced" (Palmer 2020). In 2019, there were 130 active VC firms in Poland (Krzysztofiak-Szopa et al. 2019). The majority of Polish venture capital investment comes from Polish Development Fund (PFR) Ventures, as well as from government programmes such as the National Center for Research and Development (NCBR) (Palmer 2020). It is estimated that about 52% of funds available on the market comes from the state, while 56% of Polish VCs mostly uses government support (Krzysztofiak-Szopa et al. 2019). There are just between five to seven VC teams that are experienced in managing more than two funds, while the other are younger ones, with nano-VC dominating, and have done only a few investments to date (Krzysztofiak-Szopa et al. 2019; Palmer 2020).

---

[6] eIDAS Regulation is Regulation (EU) 910/2014 on electronic identification and trust services for electronic transactions in the internal market.

For the companies in our research sample, the main capital source is own capital—see Figure 11. Only seven companies do not mention it, while three respondents refused to answer this question. In addition, business angels, individual investors and venture capital funds are other identified sources, that provided access to capital for the Polish startups.

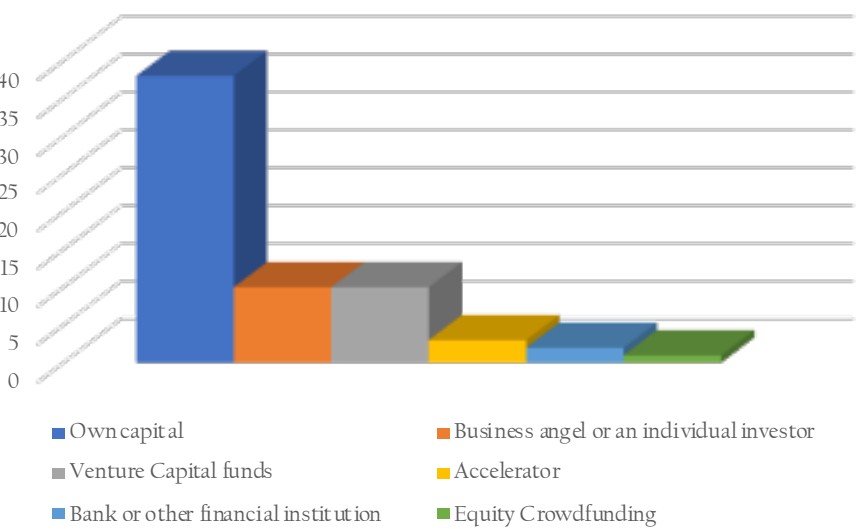

**Figure 11.** Sources of capital for Polish FinTechs.

The distribution of responses reflects the situation of the start-up financing in the Polish market. Although there are almost 350,000 angel investors in Europe, only a few hundred are present in Poland. Equity crowdfunding has not been very popular among Polish companies (in 2015 only one campaign took place), but this situation has been gradually changing (Łukowski and Zygmanowski 2019). The reason for the relatively low popularity of the alternative sources of founding among Polish companies are a lack of knowledge of equity crowdfunding by entrepreneurs from one side, and unfavourable legal conditions from the other one (Kozioł-Nadolna 2018). However, as Łukowski and Zygmanowski (2019) show, the companies from the technology and finance sectors are the early and eager adopters of this form of financing.

Although the respondents of our survey marked access to capital as less relevant risk factor, we should bear in mind that most of the FinTechs in Poland are startups or young companies, and especially during the crisis time a problem of raising capital to start a new business may appear.

*6.6. Suggestions for Policy Makers*

When we asked our respondents what kind of policy implemented by the state entities might enable their future development, the majority indicated special regulations from the Polish government (65%). Slightly fewer respondents pointed to sandboxes (56%) and just behind them tax reliefs (46%) as possible stimulus for a sector development. The survey enabled choosing more than one answer.

In Figure 12, we present the suggested changes of policy indicated by the FinTechs across the fields. What we observe, is that the special regulations have been chosen most frequently, and by each group (apart from the InsureTech that chose the option other). Regulatory sandboxes have been the second popular option chosen. This result once again confirms the need for proper legislation to enable the full development of the FinTech services.

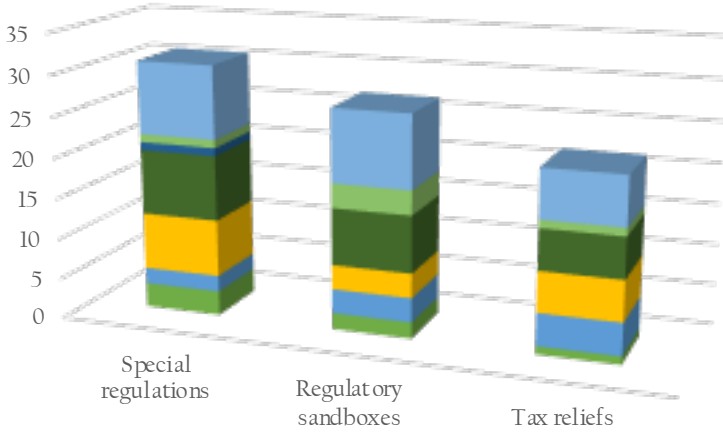

| | Special regulations | Regulatory sandboxes | Tax reliefs |
|---|---|---|---|
| ■ Payment | 9 | 9 | 6 |
| ■ Investment Management | 1 | 3 | 1 |
| ■ Insurance | 0 | 0 | 0 |
| ■ Distributed ledger | 1 | 0 | 0 |
| ■ Deposit and lending | 8 | 7 | 5 |
| ■ Banking infrastructure | 7 | 3 | 5 |
| ■ Analytics | 2 | 3 | 4 |
| ■ Accountech | 3 | 2 | 1 |

**Figure 12.** Expected government support—by field.

The companies mentioned also other kind of possible support, such as:

- Providing various sources of financing for start-ups;
- Intensifying the dialogue with developing companies from the FinTech industry;
- An active regulatory and organizational support, as well as the adoption of interesting solutions offered by FinTechs (FinTechs are en mass too small organizations to successfully promote good ideas by themselves).

It is worth to note that the above insights are in line with the KNF recommendation mentioned in Section 6.4 of this report, and the answers given to the question about the specific problems (see Table 6). Analytics, deposit, and lending, banking infrastructure, as well as investment management companies chose regulations one of the most burning issues. The mean value in these groups was equal at least 8.00, while in the payment sector it was 7.25. Only among the Accountech companies, it was relatively low (4.67). All the results show that there is a demand for clear regulations, and providing them is decisive for the future development of FinTech.

## 7. Discussion and Conclusions

In the paper, we present the stage of the development of the Polish FinTech sector and identify the main opportunities and challenges to the formation of new companies. The conclusions are formulated based on the literature review and the results of the survey run among the Polish FinTech sector in January 2020. Haddad and Hornuf (2018) documents that FinTech startups formation need not be left to chance, but active policies can influence the emergence of this new sector. Thus, the survey aimed to identify the most pressing issues for improvement.

We summarize our findings in Table 8. We refer to the factors identified in Table 1 and divide them into ones that support and pose a threat to the FinTech development in Poland.

To summarize, Poland is a fast-growing market for FinTechs. It satisfies the requirements mentioned in various studies, such as the number of secure Internet servers, mobile

telephone subscriptions, the available labor force, as well as growing tertiary education enrolment rate (Haddad and Hornuf 2018; Laidroo and Avarmaa 2019). Moreover, we observe positive trends in education, such as the constantly rising interest in IT, economics, and finance. As an opportunity for sector growth, we also recognize the fact that banks are not treated as competitors by the Polish FinTech sector. There is a collaboration that is profitable to both sides. The respondents expect that the banks will be adopting new technologies more eagerly soon, and they will modernize and digitalize. They also assume future partnerships with banks.

The results of our survey reveal, however, that there are also conditions that are not satisfied. Most importantly, the study demonstrates that regulations are the main obstacle for FinTech development. The companies consider them ambiguous, imprecise and requiring too much bureaucracy. FinTechs claimed that the rules are backward, neither follow the rapidly changing reality nor take the existence of the FinTechs into account. Another risk factor for the FinTech development is the availability of highly-skilled workers (which may stem from the fact that Poland is especially badly rated when it comes to keeping and attracting talents), and the availability of customers. Additionally, Poland lacks strong university–industry cooperation, which was underlined by our respondents in the open-answer questions. The companies mentioned the problems with access to the funds by startups as well.

**Table 8.** Factors supporting and posing the risk to FinTech formation and development in Poland.

| Category | Opportunities | Risk Factors |
|---|---|---|
| Financing system | Well developed banking system, openness to financial innovations | Venture capital financed mainly by government |
| FinTechs versus banks | Mutual cooperation | Legislation favours banks |
| Available technology | Increasing number of mobile phone subscription, and of secure internet services | Smartphones penetration below EU average, cloud servers located outside of Poland |
| Level of trust in financial incumbents | Trust in banking system above average | The overconfidence may result in the improper assessment of real risks |
| Home demand | Openness to financial innovations | Problems with finding customers |
| Education | Positive trends in education | Weak university-industry cooperation |
| Regulations | Innovation Hub, sandboxes | Regulations not sufficient for FinTechs; high level of bureaucracy |
| Knowledge in IT and financial sector | Increasing number of students in IT, finance and economics | Poland is badly rated for both keeping and attracting talents |
| IT and financial sectors productivity | Improved performance of high technology exports as a share of manufactured exports | ITC goods exports: only 2% in commercial services exports, (EU average: 7%) |
| Crisis | Increased demand for FinTechs services | Weaker opportunities for start-up financing |

Some policy implications can be formulated based on the presented analysis. First of all, there is still a need to improve current legislation to make the process more innovation-oriented. However, the regulatory changes should be preceded by the debate with the sector itself. Sandboxes are considered a good incentive for development by the majority of the FinTechs. Lastly, especially in a crisis time, tax reliefs would significantly support the sector. Moreover, the government should support the collaboration between the industry and the academic environment, since currently, it is quite scarce. Eventually, since one of the major

concerns of the FinTech sector is the possible problem with the access to qualified staff, there is a need to further support and enhance the creation of technically-oriented degree courses at the universities, to provide an adequate number of highly-skilled workers in the market.

In the end, the results of our study enable us to provide policy implications for FinTech managers. First of all, they could initiate the cooperation with universities, for instance through focused guest lectures, and in this way enhance the formation of further skilled staff (we note that the IT labour market is at the moment one of the most competitive ones, characterized by the lowest unemployment rate and the shortest time to find a job). It should also be profitable for FinTechs to sustain cooperation with banks, whom the Polish citizens trust. Eventually, to prevent home demand problems, the companies may follow the consumers' trends and adapt their offer accordingly.

Finally, we would like to outline that our research has some limitations. First of all, our sample covered 48 companies out of 233, and thus the obtained responses do not reflect the view of the whole population. The subsector of InsureTech, as well as Distributed Ledger, was underrepresented, and hence we should be aware of the possible bias when we formulate the conclusions for the whole sector. The second limitation is the fact that the survey was run just before the COVID pandemic outbreak. The environment and the demand for the FinTech services and the sentiment of the companies might have changed. Moreover, some legal solutions that FinTechs had hoped for, have been already implemented. The current situation—the crisis—is an opportunity itself for the companies from the Distributed Ledger or Payments sectors. On the other hand, the overall crisis may be harmful to others. The VC financing, which in the Polish case has been mostly supported by the government, may be limited in the future. It would restrain the development of new companies. The FinTechs outside the payment-related sector may also suffer, especially those that support the tourism or travelling sectors. Thus, the current conditions may affect and reshape the FinTech sector in Poland.

In future research, we plan to run the next round of the survey among the FinTechs in Poland and verify how the results changed in the after-pandemic economic environment. Ideally, such surveys could be repeated in the future (also across different countries), to monitor the rapidly developing Fintech environment, support its development, successfully monitor risk sources and formulate recommendations for state authorities.

**Author Contributions:** Conceptualization: B.B.-S. and A.K.; methodology: B.B.-S. and A.K.; formal analysis: B.B.-S., A.K., A.R. and K.Ś.; investigation, B.B.-S., A.K., A.R. and K.Ś.; data curation: A.K. and A.R.; writing—original draft preparation: B.B.-S. and A.K.; writing—review and editing: B.B.-S., A.K., A.R. and K.Ś.; visualization: A.K. and A.R.; funding acquisition, A.K. All authors have read and agreed to the final version of the manuscript.

**Funding:** This research was funded by the National Science Center through the project NCN MINIATURA 2019/03/X/HS4/01025 as well as Regional Initiative for Excellence programme of the Minister of Science and Higher Education of Poland, years 2019–2022, grant No. 004/RID/2018/19, financing 3,000,000 PLN.

**Institutional Review Board Statement:** Not applicable.

**Informed Consent Statement:** Not applicable.

**Data Availability Statement:** Data is available from authors upon reaasonable request.

**Acknowledgments:** We would like to thank Quantify and Quantfin for the help with obtaining the data, as well as all the respondents of our survey. We thank anonymous Referees for detailed reviews and suggestions. Eventually, we would like to thank the participants of the (virtual) International Conference on Finance and Economic Polity (ICOFEP) 2020 for the fruitful comments and discussion.

**Conflicts of Interest:** The authors declare no conflict of interest.

## Appendix A. List of the Participants of the Survey

The FinTechs who participated in our survey were: AssetLife, BanqUP, Beesfund, Bee-Tech, BillTech, Blik, Braintri (Neontri), Brutto.pl, BSS PolandSA, Centreo, Coderion,

Comperia, Currency One, Digital Teammates, Empirica, ePortfel24.pl, Faktorama, Find Funds, FinPack, Greencash, hiPRO, HOTPAY, Identt, Jakdojade, Let's Pay (PerceptusSA), Monevia, NicePay (1PayPolandSp. zo.o.), NuDelta, Payholding, PaymentTechnology, PDU, Scanye, SCFO, SportBonus, Squaber, Star Funds, Storyous, Straal, Taxxo (Columb Technologies S.A.), Tpay (Ferbuy), Trefix, Urban.one, Velochron, VoiceLab, WWWASH, Ybanking, zbiletem.pl, zrzutka.pl.

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
