# Peer review of "Triggers and Obstacles to the Development of the FinTech Sector in Poland"

_risks, doi:10.3390/risks9020030_

Round 1

Reviewer 1 Report

This study is devoted to a topical issue - development of FinTechs in Poland. The majority of studies on FinTech analyse the developments in the big FinTech markets, but it is also important to analyse the smaller markets as well. FinTechs in the Central and Eastern Europe have a potential to become important FinTech market players on the international and regional level, as the size of the local market is not decisive in case of FinTechs. 

The study aims to describe the FinTech environment in Poland, show the opportunities for its development and identify the most important threats. 

Nevertheless, there are some issues that could be improved in the paper:

-please specify the research methods

-please explain the relationship sign (positive/negative) in the Table 1

-proxy used for the "regulation" factor is missing

-the factors identified in the Table 1 - the factors that support FinTech formation are not analysed in the empirical part of the paper (in the context of the FinTechs in Poland), therefore it seems like the theoretical part is not related to the empirical part of the paper. In this case you could extend the literature review to analyse the factors that influence the evolution/development of Fintechs.

The evolution or the gradual development of Fintechs is not clearly shown.

Author Response

Thank you very much for the review of our paper. The responses to your comments are in the file attached - in the section: Responses to Comments of Reviewer 1.

Reviewer 2 Report

The paper a very interesting topic, regarding the evolution of Fintech services in Poland. 

I started reading this article with interest, which I find very interesting, but unfortunately this study is an update of the previous report made by the authors, published in May 2020 with title FinTechs in Poland: Insights, Trends and Perspectives.

Sections 3, 4, 5 and 6 are based on the previous report. Even if the source of auto-plagiarism is mentioned, most tables and figures are taken from the report, which is already published.

Which is the additional contribution compared to the previous study?

Also, in Reference section the source of report is not specified.

In my opinion, in order to accept the article for publication, several additional contributions should be included compared to the first version of the study.

Author Response

Thank you very much for the review of our paper. The responses to your comments are in the file attached - in the section: Responses to Comments of Reviewer 2.

Reviewer 3 Report

As Fintech services are projected to shape the future of the financial industry, it is crucial to investigate the drivers and obstacles for their development. Thus, the paper addresses an important question and offers some relevant policy recommendations for the development of the sector in Poland. However, explanations of methodological choices and research design as well as the possible limitations of the paper lack sufficient detail. Also, contribution of the paper to the literature on Fintech, aside from the benefits to the sector in Poland, remains vague.

General comments to the authors:

  1. Highlight your contribution to Fintech literature more clearly in the introduction. It is not sufficient to bring out the value of the study for the industry in Poland.
  2. Please explain to what extent your results are generalizable to other countries.
  3. Consider rephrasing the title of the paper in order to reflect the focus on the drivers and obstacles of the development of the Fintech sector as opposed to its evolvement over time.
  4. The title of the second chapter “state of the art” needs to be revised in order to reflect the contents of the chapter more explicitly.
  5. It is not clear if your definition of Fintech includes traditional financial institutions since they, too, offer automated financial services. If yes, you need to explain why those were excluded from the survey population.
  6. What are the limitations of your research? What are areas for future research? Please add to the conclusion.

    Remarks regarding research design

    1. I miss justification why the selected method is the preferred approach for investigating the chosen research question.
    2. The full reference to the source presenting your survey is missing. (Kliber A, Bedowska-Sójka B, Rutkowska A, Swierczynska K, Zdunkiewicz W (2020) FinTechs in Poland: Insights, Trends and Perspectives, Technical Report. Tech. Rep.) Where was it published? By what institution? Since the survey is the main component of the current paper, it is essential to explain the major details of the survey in the text.
    3. Please explain how the representativeness of the sample was tested.
    4. Please explain what your survey questions were based on. Do you use a previous survey from literature or is it a newly developed one?
    5. Please explain if there were any possible biases in the study and how they were mitigated.
    6. Figure 8 – the reviewer is not convinced that the offered responses are triggers and not outcomes of Fintech development. Please explain how the questions were derived.
    7. Figure 11 – unclear what is the basis for categories of capital. What is meant by equity capital (since several other presented items also belong to equity)? What are conclusions from the responses on the type of capital employed?

    Other comments

    1. Please specify what is meant by “non-banking Fintech sector” in section 4.
    2. Relatively outdated data on education is used on Figure 6 (ending with 2016), please update.
    3. Is it relevant to include statistics covering all fields of education? It would make sense to focus on education in the fields related to Fintech (economics, business, IT, technology?). While it might be harder to find suitable proxies in cross-country studies, you are able to take a deeper look in the context of your single country of interest.

    Grammar and formatting

    1. Some minor typing errors were found, such as “30th rang in the world” in section 2.1
    2. In the middle of a sentence the name of the referred author does not need to be in brackets: for instance, instead of (Cojoianu et al. 2020) you need to use Cojoianu et al. (2020)
    3. List of survey participants in Section 3 needs to be moved to appendix, no need to include it in the main text
    4. Formats of the charts and tables vary, please apply a unified style in line with the guidelines of the journal.
    5. Table 4: it is not clear what currency is used.
    6. In bibliography “Timaste” needs to be spelled “Tirmaste”.
    7. Figure 3. No need to have the scale running up to 120%

Author Response

Thank you very much for the review of our paper. The responses to your comments are in the file attached - in the section: Responses to Comments of Reviewer 3.

Reviewer 4 Report

The topic of the paper is very interesting, especially now when the World is in process of digitalization. Therefore, the FinTech environment provides new financial products through modern technologies. The paper is based on literature review and a survey among the FinTechs companies in Poland. The authors found that the main source of risk to the development of FinTech sector in Poland is the uncertainty about the availability of skilled workers and unclear regulations. Although, the paper has some strengths and weaknesses. The strenghts are identification of the main obstacles that pose risk to the FinTech development in Poland and formulation of policy implications. On other side, the weaknesses are that the analysis in done only in Poland and there is no comparison with other countries and short time period for a survey. In order to improve the paper, authors should explain the methodlogical part in more detail way and to recommend further research.

Author Response

Thank you very much for the review of our paper. The responses to your comments are in the file attached - in the section: Responses to Comments of Reviewer 4.

Round 2

Reviewer 2 Report

The paper is improved. 

In section Conclusion need to include the areas for future research correlated with the presentation from the introduction. Also the implications of the study results on the managerial strategies of fintech companies can be highlighted.

Author Response

At the beginning we would like to thank the reviewer for reading the second version of our paper and provide additional valuable comments. Below, we address all of them:

In section Conclusion need to include the areas for future research correlated with the presentation from the introduction.

We include areas of future research  in the Conclusions - line 708. The latter correspond to the lines 76-81 in the Introduction.

Also the implications of the study results on the managerial strategies of fintech companies can be highlighted.

We formulated the implications for managers in lines 688-694.

Reviewer 3 Report

Hi!

I am glad to see that there have improvements been made following the recommendations of the reviewers. There are a few items that I would like to call your attention to:

  1. In introduction you mention that "We adopt a qualitative approach in this study". It is unclear what you mean by "qualitative" as no qualitative methods seem to be applied in your paper. Please change or clarify in your text. 
  2. The title reflects contents of the paper better now. The phrase "risk sources" feels a bit clumsy, take a look at literature and see if there is anything that flows better. 
  3. As requested by the reviewers, please highlight areas for future research in the conclusion.
  4. Please take a look at the reference to the below article and specify the item in the list of bibliography. Where has it been published? I was not able to find it in public sources.

Laidroo L, Koroleva E, Kliber A, Rupeika-Apoga R, Grigaliuniene Z (2020) Business models of FinTechs – difference in similarity? Working Paper

5. Formatting of the paper could be further improved. Charts seem to be fuzzy and in a different font than the rest of the paper. Readability of Table 5 could be enhanced by stating numbers in thousands (one decimal after comma). 

6. There are still some typing errors that need be corrected. For instance“30th rang in the world”, "pose a thread to the FinTech development in Poland"

Good luck!

Author Response

First of all, we would like to thank the Reviewer once again for reading our article and providing further comments. Below, we respond to each of them:

1. In introduction you mention that "We adopt a qualitative approach in this study". It is unclear what you mean by "qualitative" as no qualitative methods seem to be applied in your paper. Please change or clarify in your text.

The authors decided to remove the term "qualitative" to avoid any unnecessary lack of clarity.

2. The title reflects contents of the paper better now. The phrase "risk sources" feels a bit clumsy, take a look at literature and see if there is anything that flows better.

The authors decided to change the title of the paper to the following: "Triggers and obstacles to FinTech development in Poland".

3. As requested by the reviewers, please highlight areas for future research in the conclusion.

The area for future research is mentioned in the line 81 in the Introduction. We also added the respective comments in the Conclusion section in line 708.

4. Please take a look at the reference to the below article and specify the item in the list of bibliography. Where has it been published? I was not able to find it in public sources.

Laidroo L, Koroleva E, Kliber A, Rupeika-Apoga R, Grigaliuniene Z (2020) Business models of FinTechs – difference in similarity? Working Paper

The paper is currently under review and available upon request. We added the respective note in the references. Yet, if the reviewer or editors decide that it is not enough, we can remove it from the reference list.

5. Formatting of the paper could be further improved. Charts seem to be fuzzy and in a different font than the rest of the paper. Readability of Table 5 could be enhanced by stating numbers in thousands (one decimal after comma).

The table has been altered - we added the coma to separate the thousands. The authors altered the charts by by customizing the font and the format of the graphs (from jpg to eps) - at the moment they should be displayed better.

6. There are still some typing errors that need be corrected. For instance“30th rang in the world”, "pose a thread to the FinTech development in Poland"

The Authors conducted additional proof reading of the text in order to remove typing errors.